# A New Deep-Learning-Based Model for Breast Cancer Diagnosis from Medical Images

**DOI:** 10.3390/diagnostics13111944

**Published:** 2023-06-01

**Authors:** Salman Zakareya, Habib Izadkhah, Jaber Karimpour

**Affiliations:** 1Department of Computer Science, University of Tabriz, Tabriz 5166616471, Iran; selman33111@hotmail.com (S.Z.); karimpour@tabrizu.ac.ir (J.K.); 2Research Department of Computational Algorithms and Mathematical Models, University of Tabriz, Tabriz 5166616471, Iran

**Keywords:** medical image, breast cancer diagnoses, machine learning, deep learning, classification

## Abstract

Breast cancer is one of the most prevalent cancers among women worldwide, and early detection of the disease can be lifesaving. Detecting breast cancer early allows for treatment to begin faster, increasing the chances of a successful outcome. Machine learning helps in the early detection of breast cancer even in places where there is no access to a specialist doctor. The rapid advancement of machine learning, and particularly deep learning, leads to an increase in the medical imaging community’s interest in applying these techniques to improve the accuracy of cancer screening. Most of the data related to diseases is scarce. On the other hand, deep-learning models need much data to learn well. For this reason, the existing deep-learning models on medical images cannot work as well as other images. To overcome this limitation and improve breast cancer classification detection, inspired by two state-of-the-art deep networks, GoogLeNet and residual block, and developing several new features, this paper proposes a new deep model to classify breast cancer. Utilizing adopted granular computing, shortcut connection, two learnable activation functions instead of traditional activation functions, and an attention mechanism is expected to improve the accuracy of diagnosis and consequently decrease the load on doctors. Granular computing can improve diagnosis accuracy by capturing more detailed and fine-grained information about cancer images. The proposed model’s superiority is demonstrated by comparing it to several state-of-the-art deep models and existing works using two case studies. The proposed model achieved an accuracy of 93% and 95% on ultrasound images and breast histopathology images, respectively.

## 1. Introduction

Breast cancer is the most commonly diagnosed form of cancer worldwide and the second leading cause of cancer-related deaths. In 2020, breast cancer was diagnosed in 2.3 million women globally, resulting in 685,000 fatalities. Additionally, as of the end of 2020, 7.8 million women had received a breast cancer diagnosis within the last five years [1]. Clinical studies have demonstrated that early detection is crucial for effective treatment and can significantly improve the survival rate of breast cancer patients [2].

Computer-aided detection and diagnosis (CAD) software systems have been developed and clinically used since the 1990s to support radiologists in screening, improve predictive accuracy, and prevent misdiagnosis due to fatigue, eye strain, or lack of experience [3].

The rapid progress of machine learning in both application and efficiency, especially deep learning, has increased the interest of the medical community in using these techniques to improve the accuracy of cancer screening from images. Machine learning can play an essential role in helping medical professionals in the early detection of cancerous lesions. Despite the benefits of using these techniques, cancer screening is associated with a high risk of false positives and false negatives. However, early detection of cancer can contribute to up to a 40% decrease in the mortality rate [2].

Deep-learning networks employ a deeply layered architecture that enables hierarchical learning and progressive extraction of features from data, starting from simple to progressively more complex abstractions. By autonomously learning the maximum possible set of features, the deep-learning algorithm can deliver the most precise results, rendering these networks highly effective for medical image classification and the identification of features such as lesions [4,5].

The performance of convolutional neural networks (CNNs) is hindered by several challenges, and the most popular CNNs attempt to improve their performance by simply stacking convolution layers deeper and deeper. However, the significant areas in an image can vary considerably in size, making it difficult to select the appropriate kernel size for the convolution operation due to the extreme variability in the information’s location. For information that is more locally distributed, a smaller kernel is recommended, whereas a larger kernel is chosen for information distributed more widely. Very deep networks are more prone to overfitting, and gradient updates are challenging to share throughout the entire network, in addition to being computationally expensive to naively stack large convolution processes [6,7].

Granular computing is a type of computing that focuses on the use of granules, which are small, discrete pieces of knowledge that can be combined, manipulated, or analyzed to solve complex problems. It is a form of computing that is based on the idea of breaking down complex problems into smaller, more manageable pieces. Granular computing has been used in various areas, such as data mining, decision support systems, and knowledge discovery. Granular computing can be used in image classification by dividing the image into smaller regions or sub-regions, also known as granules, and extracting features from them [8]. 

This study proposes a novel deep-learning model designed to enhance the accuracy of breast cancer detection while simultaneously reducing the network’s parameters to improve training time. Inspired by GoogLeNet and the residual block, the proposed model considers both the depth and width of the network. Multiple filters of varying sizes operate at the same level, and their outputs are concatenated and transmitted to the next module. Additionally, the use of shortcut connections and two learnable activation functions, as opposed to traditional activation functions, is expected to reduce time consumption and improve diagnostic accuracy, thereby potentially alleviating the workload of medical professionals. We also propose a granular computing-based algorithm for capturing more detailed and fine-grained information about cancer images. We will apply granular computing to the dataset before starting the training process.

The paper’s contributions can be outlined as follows:The proposed model has the highest diagnostic accuracy compared to existing breast cancer methods;This paper is the first study to use the granular computing concept in disease diagnosis;Utilizing wide and depth networks, shortcut connections, and intermediate classifiers, we design a new deep network to improve the detection of breast cancer;An attention mechanism is proposed to highlight the important features in the input image, thereby resulting in improved accuracy of the classifier;Two learnable activation functions are developed and utilized instead of traditional activation functions. Learnable activation functions provide a flexible framework that can be fine-tuned during training for optimal performance on specific tasks.

The following is the structure of this work: Section 2 addresses the related works, Section 3 presents the model and implementation, Section 4 presents the results and discussion, and ultimately, the conclusion is presented.

## 2. Related Work

Breast cancer ranks among the most prevalent cancers affecting women worldwide. Early detection and accurate diagnosis are crucial factors for effective treatment and improved patient outcomes [9]. Ultrasound imaging is a widely used method for breast cancer screening and diagnosis, but it requires skilled radiologists to interpret the images accurately [10]. According to the National Breast Cancer Foundation’s 2020 report, AI has been successfully used to diagnose more than 276,000 breast cancer cases. By analyzing breast cancer images using AI, breast lumps (masses), mass segmentation, breast density, and breast cancer risk can be identified. In the majority of patients, lumps in the breast are the most common sign of breast cancer [9]; therefore, their detection is an essential step used in CAD.

A review of deep-learning applications in breast tumor diagnosis utilizing ultrasound and mammography images is provided in [11]. Moreover, the research summarizes the latest progressions in computer-aided diagnosis/detection (CAD) systems that rely on deep-learning methodologies to automatically recognize breast images, ultimately enhancing radiologists’ diagnostic precision. Remarkably, the classification process underpinning the novel deep-learning approaches has demonstrated significant usefulness and effectiveness as a screening tool for breast cancer.

Recent studies have explored the use of deep-learning techniques, particularly convolutional neural networks (CNNs), for automated breast ultrasound image classification [12,13,14,15]. These studies have shown encouraging results. Several convolutional neural networks (CNNs) models are used for breast cancer image classifications including AlexNet, VGGNet, GoogLeNet, ResNet, and Inception. In their study, the authors of [5] categorized breast lesions as either benign or malignant. They developed a CNN model to remove speckle noise from the ultrasound images and then proposed another CNN model for classifying the ultrasound images. The study [16] discriminates benign cysts from malignant masses in US images.

In the study presented in [17], various deep-learning models were employed to classify breast cancer ultrasound images based on their benign, malignant, or normal status. A dataset comprising a total of 780 images was utilized, and data augmentation and preprocessing techniques were applied. Three models were evaluated for classification. Specifically, ResNet50 achieved an accuracy of 85.4%, ResNeXt50 achieved 85.83%, and VGG16 achieved 81.11%.

The study [18] introduced a novel ensemble deep-learning-enabled clinical decision support system for the diagnosis and classification of breast cancer based on ultrasound images. The study presented an optimal multilevel thresholding-based image segmentation technique for identifying tumor-affected regions. Additionally, an ensemble of three deep-learning models was developed to extract features, and an optimal machine-learning classifier was utilized to detect breast cancer.

In the study [14], the authors proposed a system to classify breast masses into normal, benign, and malignant. Ten well-known, pre-trained CNNs classification models were compared, and the best model was Inception ResNetV2. In [19], a vector-attention network (BVA Net) was proposed to classify benign and malignant mass tumors in the breast. 

In [20], the authors proposed a CNN-based CAD system for breast ultrasound image classification (benign and malignant lesions). The study [21] developed a deep-learning model based on ResNet18 CNN architecture for breast ultrasound image classification. In addition, the study [22] compared the performance of different deep-learning models, including CNNs, recurrent neural networks (RNNs), and hybrid models, for breast cancer diagnosis on ultrasound images. In addition to binary classification, some studies have also explored multiclass classification of breast ultrasound images. For example, the study [23] proposed a CNN-based CAD system that can classify breast lesions into four categories: benign, malignant, cystic, or complex cystic-solid. The system achieved an overall accuracy of 87% on a dataset of 1000 images.

Gao et al. have devised a computer-aided diagnosis (CAD) system geared toward screening mammography readings, which demonstrated an accuracy rate of approximately 92% [24]. Similarly, in several studies [25,26], multiple convolutional neural networks (CNNs) were employed for mass detection in mammographic and ultrasound images.

The study conducted by [3] provides a comprehensive review of the techniques used for the diagnosis of breast cancer in histopathological images. The state-of-the-art machine-learning approaches employed at each stage of the diagnosis process, including traditional methods and deep-learning methods, are presented, and a comparative analysis between the different techniques is provided. The technical details of each approach and their respective advantages and disadvantages are discussed in detail. 

Lee et al. [27] conducted a study utilizing a deep-learning-based computer-aided prediction system for ultrasound (US) images. The research involved a total of 153 women with breast cancer, comprising 59 patients with lymph node metastases (LN+) and 94 patients without (LN−). Multiple machine-learning algorithms, including logistic regression, support vector machines (SVMs), XGBoost, and DenseNet, were trained and evaluated on the US image data. The study found that the DenseNet model exhibited the best performance, achieving an area under the curve (AUC) of 0.8054. This study highlights the potential of deep-learning techniques in the development of accurate and efficient prediction systems for breast cancer diagnosis using US imaging.

Sun et al. [28] conducted a study utilizing a convolutional neural network (CNN) trained and tested on ultrasound images of 169 patients. The training dataset consisted of 248 US images from 124 patients, while the testing dataset comprised 90 US images from 45 patients. The results of the study revealed a somewhat inferior performance, with an AUC of 0.72 (SD 0.08) and an accuracy of 72.6% (SD 8.4). Notably, the validation process did not include cross-validation or bootstrapping methods. These findings suggest that further research is necessary to improve the performance of CNNs in breast cancer diagnosis using ultrasound imaging.

In a study by [29], a comparison was made between convolutional neural networks (CNNs) and traditional machine-learning (ML) methods, specifically random forests, in the context of breast cancer diagnosis. The study utilized a dataset of 479 breast cancer patients, comprising 2395 breast ultrasound images. The research also focused on different regions of the ultrasound images, including intratumoral, peritumoral, and combined regions, to train and evaluate the models. The study found that CNNs outperformed random forests in all modalities (*p* < 0.05), and the combination of intratumoral and peritumoral regions provided the best result, with an AUC of 0.912 [0.834–99.0]. While confidence intervals were provided, the method used to determine them was not mentioned. These results highlight the potential of CNNs in breast cancer diagnosis using ultrasound imaging and the importance of considering different regions of the image in the analysis.

The study proposed by [30] implemented the multilevel transfer-learning (MSTL) algorithm using three pre-trained models, namely EfficientNetB2, InceptionV3, and ResNet50, along with three optimizers, which included Adam, Adagrad, and stochastic gradient descent (SGD). The study utilized 20,400 cancer cell images, 200 ultrasound images from Mendeley, and 400 from the MT-Small dataset. This approach has the potential to reduce the need for large ultrasound datasets to realize powerful deep-learning models. The results of this study demonstrate the effectiveness of the MSTL algorithm in breast cancer diagnosis using ultrasound imaging.

The study [31] presents a review of studies investigating the ability of deep-learning (DL) approaches to classify histopathological breast cancer images. The article evaluates current DL applications and approaches to classify histopathological breast cancer images based on papers published by November 2022. The study findings indicate that convolutional neural networks, as well as their hybrids, represent the most advanced DL approaches currently in use for this task. The authors of the study defined two categories of classification approaches, namely binary and multiclass solutions, in the context of DL-based classification of histopathological breast cancer images. Overall, this review provides insights into the current state of the art in DL-based classification of histopathological breast cancer images and highlights the potential of advanced DL approaches to improve the accuracy and efficacy of breast cancer diagnosis.

The study [32] proposed a breast cancer classification technique that leverages a transfer-learning approach based on the VGG16 model. To preprocess the images, a median filter was employed to eliminate speckle noise. The convolution layers and max pooling layers of the pre-trained VGG16 model were utilized as feature extractors, while a two-layer deep neural network was devised as a classifier.

The vision transformer (ViT) architecture has been proven to be advantageous in extracting long-range features and has thus been employed in various computer vision tasks. However, despite its remarkable performance in traditional vision tasks, the ViT model’s supervised training typically necessitates large datasets, thereby posing difficulties in domains where it is challenging to amass ample data, such as medical image analysis. In [33], the authors introduced an enhanced ViT architecture, denoted as ViT-Patch, and investigated its efficacy in addressing a medical image classification problem, namely, identifying malignant breast ultrasound images.

In summary, these studies showcase the capability of deep-learning techniques in automating breast image classification and underscore the significance of devising precise CAD systems to support radiologists in detecting breast cancer. The majority of current approaches employ pre-existing deep-learning architectures for detecting breast cancer. In the following, we introduce a novel architecture that surpasses all previous methods.

## 3. Methodology

Inspired by GoogLeNet [34] and residual block [35] and adding several other features, in this paper, we developed a new deep architecture for breast cancer detection from images. GoogLeNet and residual block are based on convolutional neural network (CNN) architecture. GoogLeNet is a deep convolutional neural network architecture developed by Google’s research team in 2014. It was the winner of the ImageNet Large Scale Visual Recognition Challenge (ILSVRC) in 2014 and achieved state-of-the-art performance on a variety of computer vision tasks.

The GoogLeNet architecture consists of a 22-layer deep neural network with a unique “Inception” module that enables the network to efficiently capture spatial features at different scales using parallel convolutional layers. The Inception module combines multiple convolutional filters of different sizes and concatenates their outputs, allowing the network to capture both fine-grained and high-level features. On the other hand, a residual block is a building block used in deep neural networks that helps to address the problem of vanishing gradients during training. A residual block consists of two or more stacked convolutional layers followed by a shortcut connection that bypasses these layers. The shortcut connection allows the gradient to be directly propagated to earlier layers, allowing for better optimization and deeper architectures.

This study introduces a novel deep-learning-based architecture for breast cancer detection that stands out from existing architectures in four significant aspects, resulting in superior performance.

Proposing a granular computing-based algorithm aiming to extract more detailed and fine-grained information from breast cancer images, leading to improved accuracy and performance;Utilizing wide and deep modules, shortcut connections, and intermediate classifiers simultaneously in the architecture;Designing an attention mechanism; the attention mechanism in CNNs provides a powerful tool for selectively focusing on relevant features in the input data, enabling the network to achieve better accuracy and efficiency;Designing two learnable activation functions and using them instead of traditional activation functions.

Figure 1 depicts the overall process proposed in this paper. The input of the proposed method is a breast cancer image. If the size of the images is different, they are resized to a pre-determined size. After that, the pixels of the image are normalized between [0,1]. Resizing and normalization are preprocessing steps. After the preprocessing step, we use granular computing to highlight important features of an image. The output of the previous step is used to train the proposed deep-learning model. These steps are used for all images in the dataset. In the following, we will describe each of these cases.

### 3.1. Granular Computing

Granular computing can be used in image classification by dividing the image into smaller regions or sub-regions, also known as granules, and extracting features from them. This approach can improve the accuracy of the classification task by capturing more detailed and fine-grained information about the image [8]. 

Here, we propose general steps by which granular computing can be applied to image classification in deep learning:

**Input:** An image to be classified.

**Output:** A label representing the class of the image.

**Preprocessing:** Resize the image to a fixed size and normalize the pixel values to a range between 0 and 1.

**Granulation:** Divide the image into smaller regions or sub-regions, known as granules. This can be performed by using techniques such as windowing, tiling, or segmentation.

**Feature Extraction:** For each granule, extracts features using techniques such as local binary patterns, histograms of oriented gradients, or CNN. This will result in a set of feature vectors, one for each granule.

**Feature Aggregation:** Combine the feature vectors obtained from the granules and use them to classify the image. This can be performed by using techniques such as mean pooling or max pooling.

Figure 2 shows the proposed steps for granular computing used in this paper. Considering the above steps, we propose Algorithm 1 for applying granular computing in this paper. In this algorithm, we have used the pre-trained VGG16 architecture to extract features for each granularity. The size of each granular is considered to be 32 × 32. We apply granular computing to the dataset before starting the training process.
**Algorithm 1.** Extracting more detailed features by granular computingRepeat the following steps for all imagesimg = Load the imagePreprocessing step: resizing and normalizationimg = resize(img, (224, 224))img = normalize(img/255.0)Granulation step: split the image in windows of size 24 * 24granules = []for i in range(0, 224, 32):for j in range(0, 224, 32):granule = img[i:i + 32, j:j + 32,:]granules.append(granule)Feature Extraction step:model = VGG16(weights= ‘imagenet’, include_top = False, input_shape = (32, 32, 3))features = []for granule in granules:feature = model.predict(np.expand_dims(granule, axis = 0)).squeeze()features.append(feature)Feature Aggregation step:features = np.array(features)aggregated_features = np.mean(features, axis = 0)

### 3.2. Learnable Activation Function

Sigmoid, ReLU, and tanh are widely used activation functions in artificial neural networks, and they perform satisfactorily in many cases. However, these functions have some limitations that make their use suboptimal, thus necessitating the development of learnable activation functions.

Learnable activation functions in artificial neural networks are like functions that can adapt and modify themselves based on the input received by the neural network. These functions are characterized by a group of parameters that can be fine-tuned by the neural network during the training process. The primary advantage of learnable activation functions is their flexibility and adaptability in adjusting to different types of input data being processed. There are two primary types of learnable activation functions in artificial neural networks:-Parametric activation functions: These functions have a fixed form (such as sigmoid or ReLU), but they add extra learnable parameters to them. The neural network changes these parameters to adjust the activation function appropriately on training data;-Adaptive activation functions: These functions do not have any fixed form or formula. Instead, they rely on a neural network structure such as RNN or LSTM to learn appropriate activation functions.

We, here, develop two learnable activation functions named LAF_sigmoid and LAF_relu. To develop a new parametric learnable activation function, we start by defining a general form of the function that has learnable parameters. Let us call this function “Learnable Activation Function”, or LAF for short:LAF(x; W) = a ∗ F(x; W) + b (1)

Here, *a* and *b* are adjustable parameters that are learned during training, and *F*() is a non-linear function that defines the shape of the activation function. The weight matrix *W* contains learnable values that determine the shape of *F*(), and it is optimized through backpropagation.

To further develop the LAF, we can choose a suitable non-linear function *F*(). One possible choice is the sigmoid function:F(x; W) = 1/(1 + exp(−Wx))(2)

The sigmoid function is a common choice for activation functions due to its smoothness and boundedness, which is important for the stable training of neural networks. The LAF with the Sigmoid function becomes:LAF_sigmoid(x; W, a, b) = a ∗ (1/(1 + exp(−Wx))) + b(3)

This activation function will be used in the dense layers of the network.

Another possible choice for *F*() is the ReLU function:F(x; W) = max(0, Wx) (4)

The ReLU function is preferred for some tasks because of its simplicity and computational efficiency. The learnable parameters *a* and *b* can be added to shift and scale the ReLU function, resulting in the LAF with the ReLU function:LAF_relu(x; W, a, b) = a ∗ max(0, Wx) + b(5)

This activation function will be used in the convolutional layers of the network.

The values of *a*, *b*, and *W* can be trained through backpropagation using gradient descent or other optimization algorithms. The choice of the initial values and number of hidden units are important factors that can affect the success of training the neural network using LAFs. There are several advantages of using learnable activation functions in artificial neural networks:Improved performance: By incorporating learnable activation functions, the neural network performance can be improved significantly. This is because the activation function adapts to the input data, allowing for a more accurate representation of complex relationships between features;Non-linear mapping: Learnable activation functions allow for non-linear mappings between input and output, which can capture more complex patterns in the data;Flexibility: With traditional activation functions, the network architecture is fixed. However, using learnable activation functions allows for more flexibility in the network architecture, as the activation function can be modified according to the specific task;Reduced overfitting: Learnable activation functions can also help reduce overfitting, as they can adapt to the input data and generalize better to new data that has not been seen before;Efficient training: The use of learnable activation functions can also make the training process more efficient by allowing gradients to be propagated through the network more smoothly. This can lead to faster convergence and improved performance.

### 3.3. The Attention Mechanism

The attention mechanism in convolutional neural networks (CNNs) is a tool that enables networks to selectively focus on specific parts of input data that are crucial for making decisions. Mimicking the process of human attention, this mechanism allows neural networks to attend selectively to relevant features in input data. The attention mechanism can be incorporated into various parts of a network, including the input layer, the convolutional layer, or the dense layer. Typically, the attention mechanism is added on top of the convolutional layer as an auxiliary module.

The attention mechanism takes as input the output features of the preceding layer and computes a set of attention weights for each feature. These weights reflect the importance of each feature for the current task. To compute these attention weights, a sub-network called the attention module, which comprises one or more layers, is employed. The attention weights are then multiplied element-wise with the output features of the preceding layer to obtain a weighted sum of the characteristics. This weighted sum is then passed to the next layer of the network. By doing this, the attention mechanism selectively focuses on key parts of the input data and enhances their impact on the output.

The attention mechanism can be trained end-to-end using backpropagation, and the attention weights can be learned jointly with the neural network parameters. During training, the attention mechanism learns to weigh the importance of various features based on their relevance to the task. This enables the network to selectively attend to the most informative parts of the input data and ignore irrelevant or noisy features.

In this section, we propose an attention mechanism to apply to the output layer (top layer). The attention mechanism in CNNs can highlight the salient regions in images that are significant for the classification task.

In the case of breast cancer detection, this can be useful since certain regions of the breast image may contain more relevant features for cancer detection compared to others. Here, we develop an attention mechanism for CNN:Start with a standard convolutional layer with filters of size (k, k) and stride s;Add a second convolutional layer with filters of size 1 × 1 and stride 1. This layer will compute a scalar attention weight for each pixel in the input image;Apply a Softmax activation function to the output of the attention layer to ensure that the weights sum up to 1 for each pixel;Multiply the attention weight maps element-wise with the input image to obtain the attended input image;Feed the attended input image into the next layer of the CNN.

The idea behind this attention mechanism is that the second convolutional layer learns to compute a scalar attention weight for each pixel in the input image, based on its relevance to the task at hand. The softmax activation function ensures that the attention weights sum up to one for each pixel, making them interpretable as a probability distribution over the pixels. The element-wise multiplication of the attention weight maps and input image highlights or downplays certain pixels, improving the accuracy of the CNN on the given task.

### 3.4. Wide and Depth Networks, Short Connections, and 1 × 1 Convolutional Layers

Our developed network takes advantage of the features of wide and deep networks, short connections, and 1 × 1 convolutional layers. In the following, we will examine each one.

Wide and deep neural networks such as GoogLeNet offer improved accuracy, higher capacity, faster convergence, and better regularization. They have demonstrated impressive performance in various tasks, including image recognition, speech recognition, and natural language processing. Their advantages make them an attractive choice when designing neural networks. 

In neural networks, short connections, which are used in ResNet and DenseNet networks, are a type of connection between the neurons that bypass one or more layers in the network. These connections allow information to flow between two layers that are not directly connected in the network architecture. Short connections, also known as skip connections, in neural networks can be represented mathematically as an element-wise summation or concatenation operation between the input to a layer and the output of that layer. In other words, the output of a layer is added to or concatenated with the input to that layer or a previous layer. For example, in a convolutional neural network (CNN), a short connection can be introduced between two convolutional layers by adding the output of the first convolutional layer to the input of the second convolutional layer. This can be added as follows:x1 = Convolutional_layer_1(input)
x2 = Convolutional_layer_2(x1 + input)
where “+” denotes element-wise summation.

Short connections or residual connections in neural networks have several advantages:Improved gradient flow: By adding short connections, the gradient can flow through the network more effectively, which eliminates the vanishing gradient problem. The gradient can be propagated directly to earlier layers, allowing the network to train deeper architectures;Improved training speed: The use of short connections reduces the number of layers in the critical learning path, which can speed up the training process. The reduced depth also means that less computation is required, resulting in a more efficient model;Improved accuracy: Short connections enable the learning of more complex functions by allowing the network to make use of the information from earlier layers. This can result in higher accuracy in tasks such as image recognition and speech processing;Reduced overfitting: Short connections can help reduce overfitting by providing a regularization mechanism. They allow the network to learn simpler representations for the input data, which leads to better generalization.

In CNNs, 1 × 1 convolutional layers are utilized as a type of layer that executes a convolution operation by convolving the input tensor with a kernel of size 1 × 1. Despite their small size, 1 × 1 convolutional layers have various advantages in CNNs:Dimensionality reduction: 1 × 1 convolutional layers can be used to reduce the dimensionality of feature maps, which can be useful in reducing the computational complexity of CNNs while maintaining their accuracy. By using 1 × 1 convolutional layers, the number of parameters can be reduced while still retaining the important features;Non-linear transformations: Even though it has a kernel of size 1 × 1, this layer applies non-linear transformations to the input feature maps. The non-linear activation function applied after the convolution operation contributes to this non-linearity;Improved model efficiency: By reducing the number of parameters, 1 × 1 convolutional layers reduce the computational cost of the model. This can, in turn, improve the efficiency of the implementation of the model, allowing it to be run on smaller devices or with fewer computational resources;Feature interaction: A 1 × 1 convolutional layer can act as a feature interaction layer and induce correlations between features, which can further enhance the representation power of the network.

These advantages make 1 × 1 convolutional layers an important building block in CNNs, especially in deeper networks where computational cost and memory usage are of key concern.

### 3.5. The Designed Architecture

To reduce the high volume of processing in the GoogLeNet network, we changed the architecture from fully connected to sparsely connected network architectures within the convoluted layers. The Inception layer, which was inspired by the Hebbian principle of human learning, is critical to this sparsely connected architecture. For example, a deep-learning model for recognizing a particular pattern (e.g., face) in an image might have a layer that focuses on individual parts of an image. The next layer then focuses on the overall pattern in the image and identifies the various objects in it. To this end, the layer requires appropriate filter sizes to detect these objects. The Inception layer is crucial in this scenario, allowing internal layers to determine which filter size is relevant for learning the required information. Therefore, even if the size of the pattern in the image is different, the layer can recognize it accordingly. 

The proposed system is shown in Figure 3, which shows the block diagram used to diagnose medical images. The proposed system has the same structure as the simple GoogLeNet network. We replaced the Inception module with a new module named X-module. The structure of the new deep neural network consists of the following:

Input layer;A convolutional-based attention layer;Convolution layer;Two X modules with different filter sizes followed by a down-sample module;An auxiliary classifier with a learnable Softmax classifier;Three X modules with different filter sizes followed by a down-sample module;An auxiliary classifier with a learnable Softmax classifier;An X-module followed the Average pool, dropout layer;A dense layer-based attention layer;Learnable Softmax classifier as output layer.

The details are explained as follows:

Input Layer: In this step, the medical image is entered into the system.

Convolutional-based attention layer: This layer allows for the selective focus on specific parts of the input data that hold significance in determining an outcome. 

Convolution Layer: This layer uses convolution operations to produce new feature maps. 

X-Module: This module considers both the depth and width of the network, with multiple filters of varying sizes operating at the same level. The outputs of these filters are concatenated before being transmitted to the subsequent module. The main unit in X-module is a sub-block called R-block (Figure 4), which is inspired by the residual block. The main difference is that the designed block uses learnable activation functions. The block uses a shortcut connection; the input is added to the output of the block to pass gradient updates through the entire network easily and reduce overfitting. The R-block consists of two convolutional layers stacked on top of each other, with the first layer being succeeded by a batch normalization layer and a learnable activation function that is dependent on parameters. Using a parameter learnable activation function helps to reduce time consumption and better learning. Three types of R-blocks are implemented upon the filter size. The filter size of the two convolutional layers in the first R-block is 3 × 3. In the second R-block, the filter size of the two convolutional layers is 5 × 5. The first convolutional layer in the third R-block has a 3 × 3 filter size and the second convolutional layer has a 5 × 5 filter size. We have reduced the number of parameters and computational costs in the X-module by incorporating an additional 1 × 1 convolution in the initial layer, preceding the 3 × 3 and 5 × 5 convolutions. An extra 1 × 1 convolution is also utilized after the max pooling layer. 

Different configurations of the X-module are implemented to build different deep neural network (DNN) models. In the first X-module, the first R-block is 3 × 3, the second is 5 × 5, and the third R-block has a 3 × 5 convolution filter size. In the second model, three R-blocks of filter sizes 3 × 3 are utilized. In the third model, three R-blocks of filter sizes 5 × 5 are utilized. The structure of the R-block and X-module are shown in Figure 4 and Figure 5, respectively.

Downsampling module: Following the X-module, this module is implemented to decrease both the size of the feature map and the number of network parameters. The pooling max function is concatenated with a 3 × 3 convolutional layer. As stated before, deep neural networks are computationally expensive. To make it cheaper, the number of input channels is limited by adding a 1 × 1 convolution before the 3 × 3 convolution.

Dense layer-based attention layer: This attention takes as input a 3D tensor representing the output features of the previous layer and outputs a 2D tensor of attention scores, where each score represents the relevance of a specific feature.

Output layer: The output in this step will be normal or abnormal.

## 4. Result and Discussion

The experiments were conducted using the Cairo University ultrasound images dataset and the breast histopathology images dataset for training and testing. The code was written in Python, and the experiments were performed on Kaggle, leveraging the power of their hardware, including GPUs. To train the model, the following settings were employed: Adam optimizer with a learning rate set to 0.0001, 100 epochs, a batch size of 32, and a dropout rate of 50%. The number of epochs in a CNN is one of the hyperparameters that can be tuned to improve the performance of the model. The number of epochs refers to the number of times the entire training dataset is passed through the CNN during the training phase. To tune the number of epochs, we used the early stopping technique. Early stopping is a method used to prevent overfitting by monitoring the validation loss during training and stopping the training process once the validation loss starts to increase. After applying this technique, we set the number of epochs to 100. 

In this study, the proposed approach is assessed in terms of several performance metrics, including accuracy, loss, precision, recall, and F1 score. These metrics are defined as follows:

Accuracy: This metric measures the overall performance of the model by calculating the percentage of correctly predicted labels to the total number of samples in the test dataset. Mathematically, it can be expressed as:Accuracy = (Number of Correct Predictions)/(Total Number of Predictions) (6)

Loss: This metric represents the error between the predicted output and the actual output. The loss function is typically defined during the training phase of the model, and it is used to optimize the model parameters by minimizing the difference between its predictions and the true values. The most commonly used loss function in deep learning is the mean squared error (MSE), which measures the average of the squared differences between the predicted and true values.

Precision: This metric measures the proportion of true positives (samples that were correctly classified as positive) to the total number of positive predictions made by the model. It can be calculated as:Precision = True Positives/(True Positives + False Positives) (7)

Recall: This metric measures the proportion of true positives to the total number of true positives and false negatives in the dataset.

F1: This metric calculates the harmonic mean of precision and recall.

Different DNN models are implemented using different configurations of the X-module, as depicted in Figure 5. The first model has three R-blocks of 3 × 3 convolution filters. The second model utilized three R-blocks of 5 × 5. In the third model, the first R-block is 3 × 3, the second is 5 × 5, and the third R-block has a 3 × 3 filter size in the first convolutional layer, and the second convolutional layer has a 5 × 5 filter size. We have utilized various filters and kernels (kernel size). By specifying multiple values for the kernel parameter within a filter, our model can effectively identify patterns that occur at different scales within an image. The incorporation of multiple kernels also assists in reducing overfitting and improving the generalization of the model. This is because including filters with varying kernel sizes compels the network to learn more diverse and robust feature representations, leading to an improved ability to generalize the model to new images.

The Cairo University ultrasound images dataset was collected in 2018 that consists of 780 images with an average image size of 500 × 500 pixels [36]. The images are categorized into three classes, which are normal (133 images), benign (487 images), and malignant (210 images). The data collected at baseline include breast ultrasound images among women in ages between 25 and 75 years old. The number of patients is 600 female patients. Because the number of images in different classes is unbalanced, this may cause the model to learn some classes better than others, and this can cause the model to perform inappropriately during use or testing. To prevent this from happening, we randomly selected an equal number of images from each class. Before starting the training process of the model, due to the lack of data, we start the data augmentation process. We resize the dataset using cubic interpolation to fit the input requirements of the model. For augmentation, we applied width and height shifts of 0.1 and a horizontal flip, which tripled the size of the dataset. With this technique, we tripled the number of data for each class. After this process, we split the dataset into training and test sets, allocating approximately 80% for training and 20% for testing. Of course, we have not used these new images for testing. We maintain the sequence of each image so that every image appears only once in each of the aforementioned sets.

Figure 6 depicts the accuracy and loss diagrams for the three proposed models on the Cairo University ultrasound dataset.

On the Cairo University ultrasound images dataset, the performance of the three proposed CNN models in the testing phase is summarized in Table 1. The third model, DNN R3_R5_R35, which uses a different size of convolutional filters, achieves the best accuracy and low loss. It achieved 93% accuracy performance. 

Table 2 is the resultant confusion matrix of the DNN R3_R5_R35 model. This table shows promising results so that the proposed model has correctly diagnosed the presence or absence of cancer in most cases, and it has not been able to correctly diagnose only five cases out of 68 cases.

As we used some of the features of the state-of-the-art GoogLeNet and ResNet architectures in the design of the new architecture, we compared the proposed architecture, i.e., DNN R3_R5_R35, with these architectures in Table 3. To perform this comparison, we have utilized GoogLeNet with 22 layers and ResNet with 50 layers for comparison. The results indicate that ResNet outperformed GoogLeNet in two critical evaluation metrics: accuracy and F1 score. Inspection of the table containing the results reveals that the proposed method has surpassed both of these models and yielded a higher detection accuracy than these two state-of-the-art architectures.

The performance of GoogLeNet during the training phase is shown in Figure 7. A comparison between Figure 6 and Figure 7 demonstrates that there is a suitable performance for the proposed models and no overfitting compared to GoogLeNet. The DNN R3_R5_R35 model that uses different sizes of convolutional filters achieves the best performance. Here, we compare the proposed architecture with the state-of-the-art image processing architectures in terms of prediction accuracy and loss on test data. Table 4 shows these comparisons. The proposed model is superior to all existing image processing architectures in terms of prediction accuracy on breast cancer images.

We also applied the proposed model to the breast histopathology images dataset to further evaluate it. The original dataset consisted of 162 whole-mount slide images of breast cancer specimens scanned at 40×. From that, 277,524 patches of size 50 × 50 were extracted (198,738 IDC-negative and 78,786 IDC-positive). Invasive Ductal Carcinoma (IDC) is the most common subtype of all breast cancers. We choose 40,000 images in total from the dataset: 20,000 random images from both classes. We split the dataset into training and test sets, allocating 80% for training and 20% for testing. The comparison results of the proposed DNN R3_R5_R35 model against existing approaches are listed in Table 5. The best results are highlighted in bold. It is evident from the table that the proposed model has outperformed the other models in the two assessed criteria.

The objective of this study was to enhance the accuracy of breast cancer detection through the application of deep-learning techniques in the development of computer-aided detection systems. The proposed model, utilizing various filter sizes, demonstrated 93% and 95% accuracy on two distinct datasets: ultrasound images and breast histopathology images, respectively. The second goal was to decrease the parameters of the network, aiming to improve the training time. The time issue during the training process for any deep-learning model is still challenging and depends on the facilities that be used. Training the model on ultrasound and histopathology images takes less than two hours and less than six hours, respectively, which is suitable compared to other DNN models. 

A short review above, we can conclude the main findings of this paper as follows:The granular computing technique used in this paper, by breaking down images into smaller, more granular components, can effectively extract features from images, allowing for more accurate and efficient image analysis. This leads to increased efficiency by reducing the computational complexity of image analysis tasks. Moreover, breaking down images into smaller, granular computing can improve the accuracy of image analysis tasks, leading to more reliable results;Activation functions with learnable parameters offer greater flexibility and adaptability compared to traditional activation functions with fixed parameters. This allows the network to better adapt to different types of data and tasks. These functions can also improve the flow of gradients through the network during training, making it easier to optimize the network and reduce the risk of vanishing gradients. Better regularization is another advantage of these functions. Learnable activation functions can be used as a form of regularization, helping to prevent overfitting by constraining the network’s capacity and reducing the risk of memorization;In this study, a range of filters and kernels of varying sizes were employed to effectively identify patterns at multiple scales within an image. By incorporating multiple kernels within a filter, the network was able to learn diverse and robust feature representations, which helped to reduce overfitting and improve the generalization of the model. This approach enabled the model to consider a wider range of input features, leading to higher accuracy in complex tasks compared with a model that employs a single filter and kernel. The use of multiple kernels within a filter, therefore, represents an effective strategy for improving the ability of a neural network to generalize to new images by facilitating the learning of more sophisticated features across a range of spatial scales;Utilizing a wide and depth network, shortcut connections, attention layers, auxiliary classifiers, and using a learnable activation function improves the accuracy of diagnosis and consequence and decreases the load on doctors. In addition, using 1 × 1 convolutions reduces time consumption in the model. Compared to existing breast cancer methods, the proposed model achieves the highest diagnostic accuracy.

There are two potential limitations to the presented model: The extraction of some patterns from the image may be dependent on the granularity size. In the proposed granulation, the granularity size was set to 32 × 32 pixels, regardless of the image size. Consequently, some patterns may not be extracted, weakening the effectiveness of granulation. However, the model’s overall performance demonstrates that the proposed granulation method outperformed state-of-the-art models for the datasets under consideration;Incorporating granularity in a model requires additional time before the training process can commence. It is worth noting, however, that once these granules have been established, they can be reused multiple times.

## 5. Conclusions and Future Work

By automating the diagnosis process, healthcare professionals can focus on providing personalized treatment plans for patients. This not only improves patient outcomes but also reduces healthcare costs by minimizing unnecessary procedures and tests. The use of machine-learning and deep-learning techniques in breast cancer detection has the potential to revolutionize the way breast cancer is diagnosed and managed. Incorporating these tools into healthcare systems has the potential to lower cancer-related mortality rates and enhance the overall management of breast cancer. This paper proposed a novel granular computing-based deep-learning model for breast cancer detection, which is evaluated under ultrasound images and breast histopathology images datasets. The proposed model has used some effective features of GoogLeNet and ResNet architectures (such as wide and depth modules, 1 × 1 convolutional filters, auxiliary classifiers, and skip connection) and has added some new features such as granular computing, activation functions with learnable parameters, and attention layer to the new architecture. Granular computing can extract the important features of the image and create a new image with the important features highlighted before sending it to the training process. This feature makes the model require fewer images than in the case where granularity is not used. The proposed model achieved an accuracy improvement compared to state-of-the-art models. In particular, the deep-learning model based on granular computing exhibited an accuracy of 93% and 95% on two real-world datasets, ultrasound images and breast histopathology images, respectively. The model delivered promising results on the datasets. The findings may encourage radiologists and physicians to leverage the model in the early detection of breast cancer, leading to improved diagnosis accuracy, reduced time consumption, and eased workload of doctors. It has been confirmed that granular computing has a positive effect on the performance of problems where the number of available images is small, such as breast cancer.

For future work, the following items can be performed: (1) The framework can be further optimized for real-time performance. Taking inspiration from the MobileNet architecture, e.g., separable convolutions feature, the number of parameters of the proposed architecture can be reduced so that the accuracy does not decrease much. (2) Instead of granular computing, one can use fuzzy clustering. This method uses fuzzy logic to group pixels in an image into clusters based on their similarity. (3) There are usually very few medical photos. The number of these photos can be increased with techniques such as Sketch2Photo [42] before starting the learning process. This will increase the accuracy of the model. (4) Over the last 30 years, hyperspectral imagery (HSI) has gained prominence for its ability to discern anomalies from natural ground objects based on their spectral characteristics. The importance of HSI has been recognized in a variety of remote sensing applications, including but not limited to object classification, hyperspectral unmixing, anomaly detection, and change detection [43]. We can use this technique to identify breast cancer. (5) The primary challenge in content-based image retrieval (CBIR) systems is the presence of a semantic gap that must be narrowed for effective retrieval. To address this issue, various techniques, such as those outlined in [44], can be employed to incorporate semantic considerations. (6) To reduce the processing time, it is suggested to use distributed and parallel similarity retrieval techniques, such as [45], on large CT image sequences. (7) The proposed framework can be extended to include other types of cancer detection, such as lung or prostate cancer. This would enable the development of a comprehensive cancer detection system that can be integrated into existing healthcare systems. 

## Figures and Tables

**Figure 1 diagnostics-13-01944-f001:**
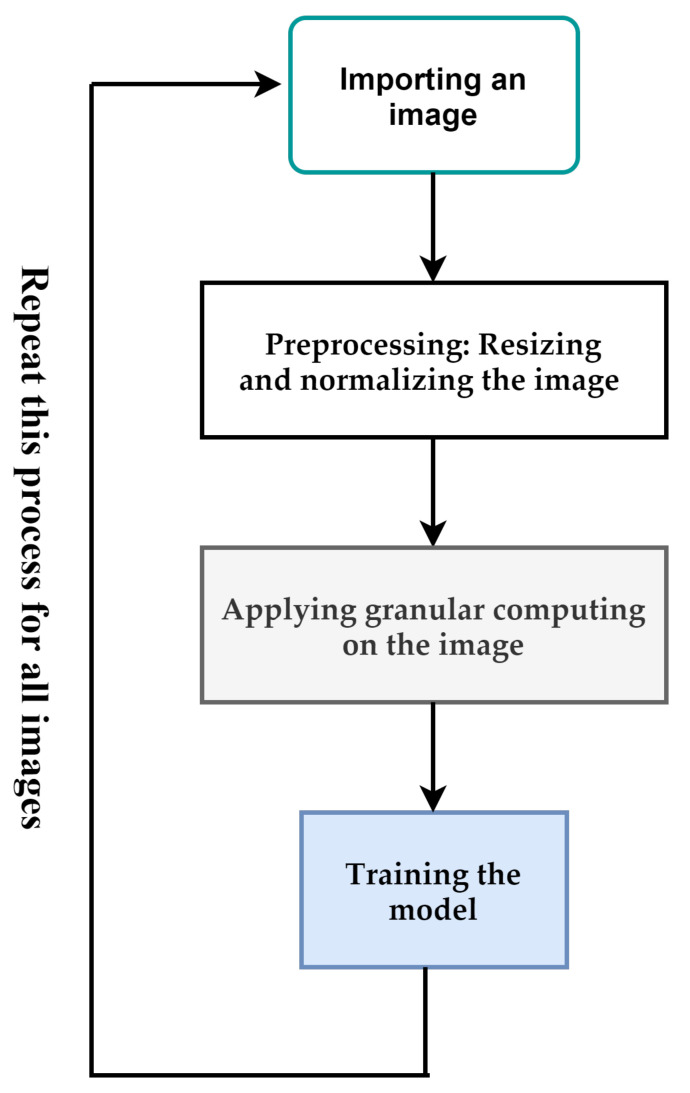
The overall process of the proposed method. All steps in this figure are repeated for all images.

**Figure 2 diagnostics-13-01944-f002:**
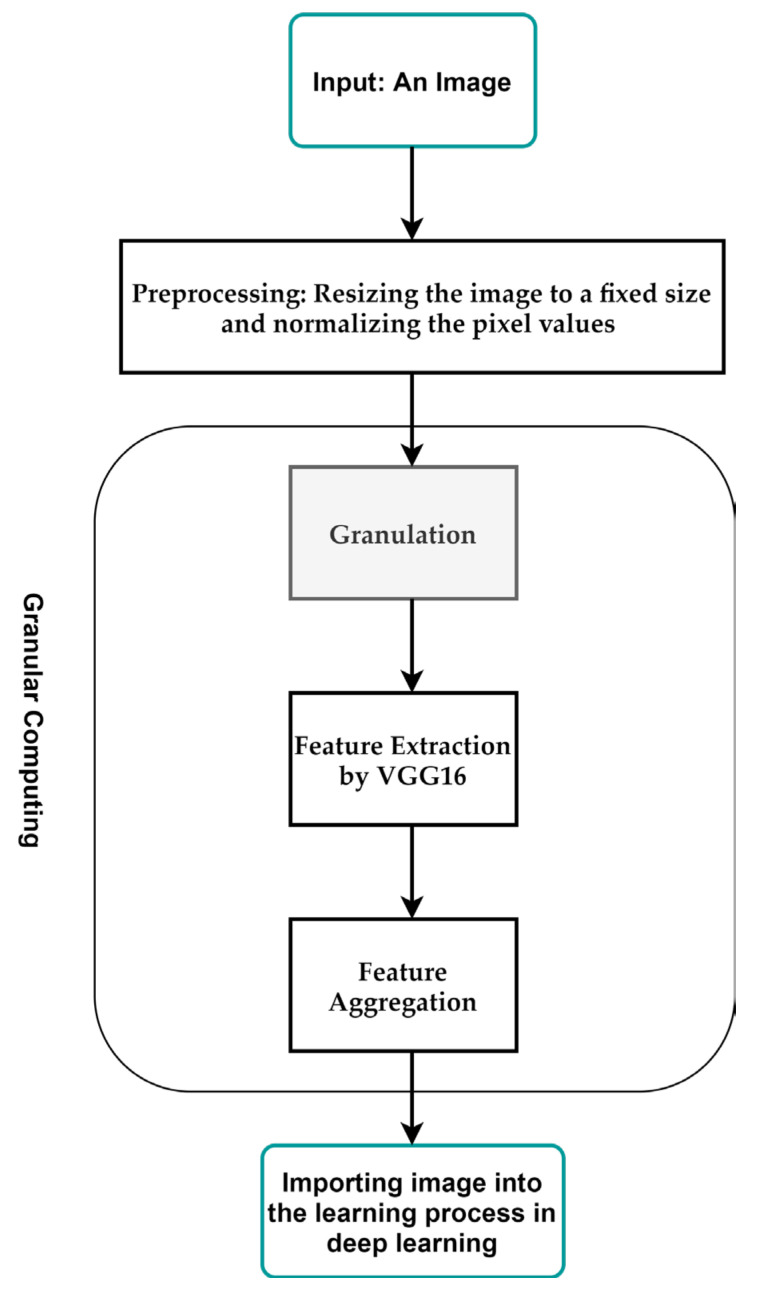
The granular computing process proposed in this paper.

**Figure 3 diagnostics-13-01944-f003:**
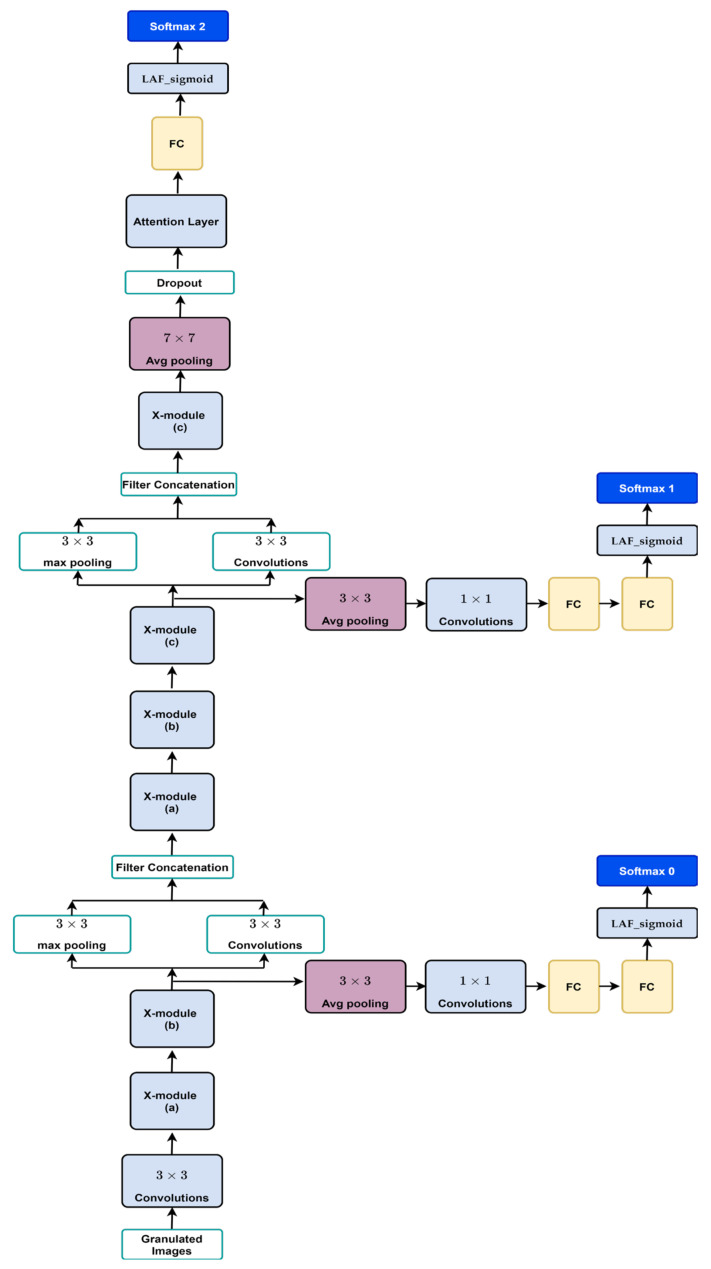
The new CNN system proposed to diagnose medical images.

**Figure 4 diagnostics-13-01944-f004:**
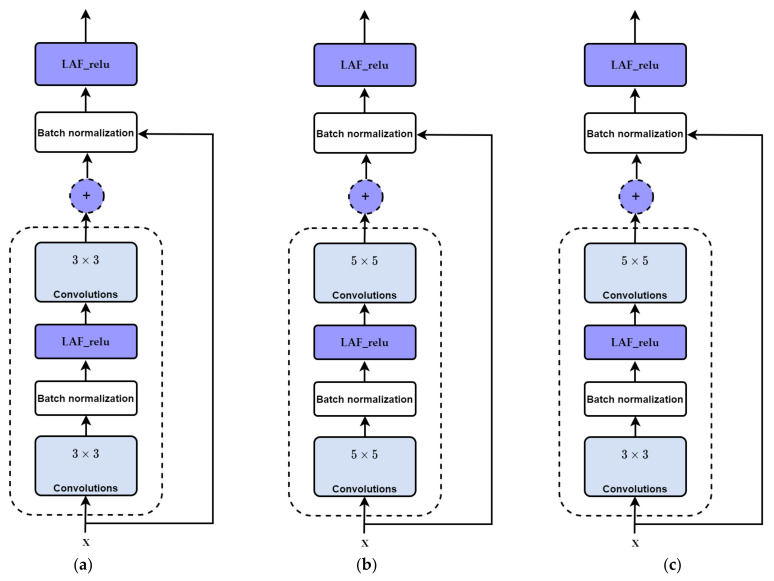
Structure of R-block (the main units of X-module): (**a**) R-block 3 × 3. (**b**) R-block 5 × 5. (**c**) R-block 3 × 5.

**Figure 5 diagnostics-13-01944-f005:**
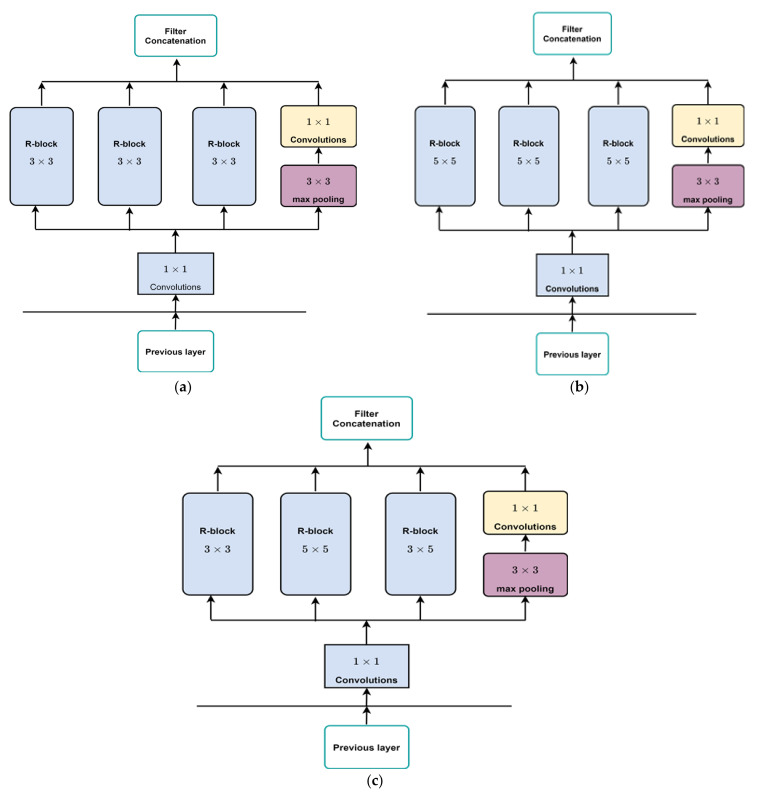
The different used structures into X-module: (**a**) X-module includes three R-blocks of 3 × 3, called DNN R3_R3_R3. (**b**) X-module includes three R-blocks of 5 × 5, called DNN R5_R5_R5. (**c**) X-module includes three R-blocks of 3 × 3, 5 × 5, and 3 × 5, called DNN R3_R5_R35.

**Figure 6 diagnostics-13-01944-f006:**
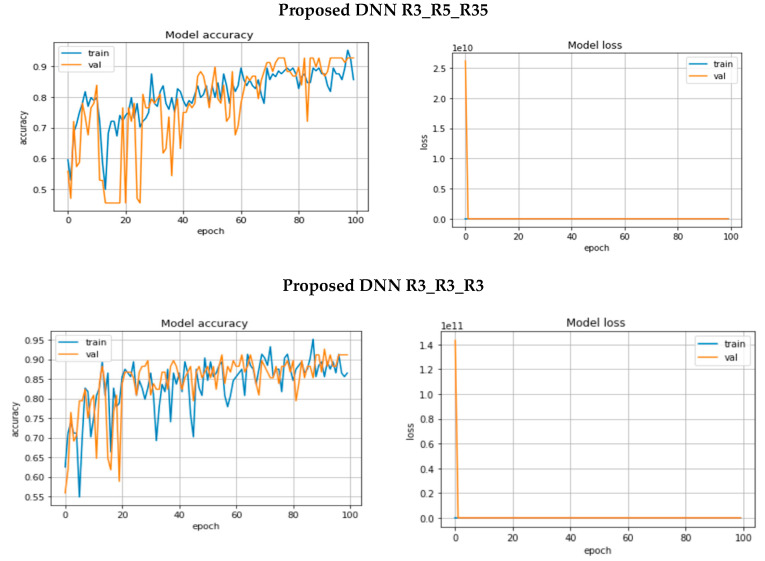
Accuracy and loss diagrams for the three proposed models.

**Figure 7 diagnostics-13-01944-f007:**
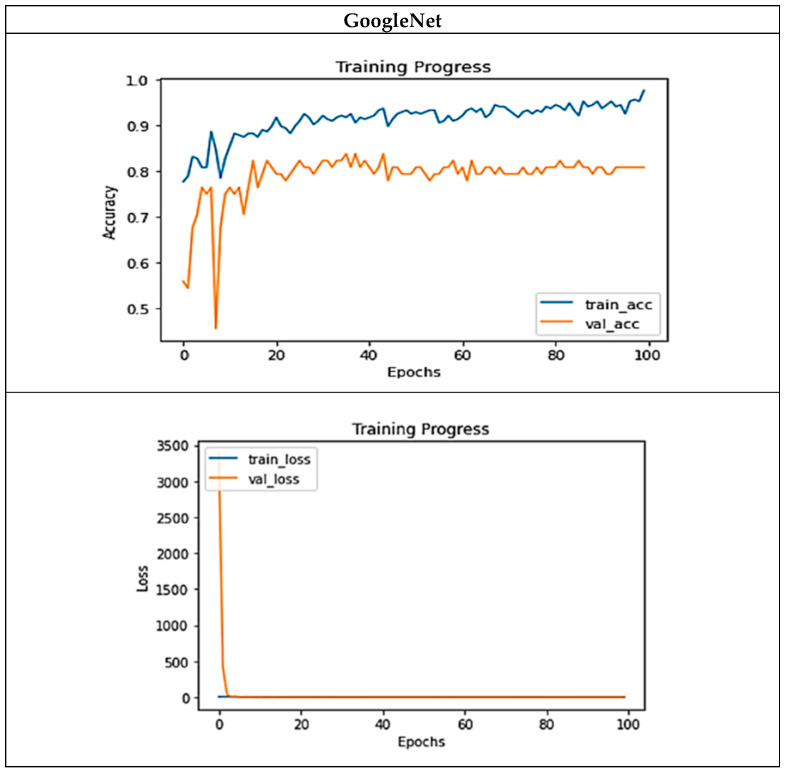
Accuracy and loss diagrams for the GoogLeNet.

**Table 1 diagnostics-13-01944-t001:** The newly developed model results on Cairo University ultrasound images dataset (test part).

Deep-Learning Model	Precision	Recall	F_1_-Score	Loss	Accuracy
Proposed DNN R5_R5_R5	0.88	0.88	0.88	0.2919	0.88
Proposed DNN R3_R3_R3	0.91	0.91	0.91	0.2445	0.91
Proposed DNN R3_R5_R35	0.93	0.93	0.93	0.2103	0.93

**Table 2 diagnostics-13-01944-t002:** Confusion matrix for the test dataset. (**0** indicating no breast cancer and **1** indicating existing breast cancer, one of the benign and malignant in the image.).

	Predicted		
**Actual**		**0**	**1**
	**0**	33	4
	**1**	1	30

**Table 3 diagnostics-13-01944-t003:** Comparison of the proposed model against GoogLeNet and ResNet on the test dataset.

Deep-Learning Model	Precision	Recall	F_1_-Score	Loss	Accuracy
GoogLeNet	0.86	0.87	0.85	0.59	0.87
ResNet50 (Residual Network)	0.87	0.85	0.86	0.51	0.88
DNN R3_R5_R35	0.93	0.93	0.93	0.2103	0.93

**Table 4 diagnostics-13-01944-t004:** Comparison of the proposed model against nine state-of-the-art image processing models on the Cairo University ultrasound images dataset.

Deep-Learning Model	Loss (%)	Accuracy (%)
AlexNet	66	69
ZFNet	64	69
VGG-16	6	73
Inception v4	46	85
MobileNet	55	85
WideResNet	39	88
GoogLeNet	59	87
ResNet34	61	83
ResNet50	51	88
Proposed DNN R3_R5_R35	**21**	**93**

**Table 5 diagnostics-13-01944-t005:** Comparative analysis on breast histopathology images dataset.

Ref.	Year	Method/Model	Accuracy (%)	F_1_-Score (%)
[32]	2023	Transfer learning with VGG16	91	89
[33]	2023	ViT-Patch 32	89	-
[37]	2021	CNN architecture	87	87
[38]	2021	DenseNet121 Model	86	87
[38]	2021	DenseNet169	85	86
[38]	2021	MobileNet	86	85
[38]	2021	ResNet50	84	83
[38]	2021	VGG19	84	80
[38]	2021	VGG16	85	85
[38]	2021	EfficientNetB0	84	84
[38]	2021	EfficientNetB4	82	82
[38]	2021	EfficientNetB5	84	84
[39]	2020	Residual learning-based CNN	84	83
[40]	2020	CNN	85	82
[41]	2021	Patch-Based Deep-Learning Modeling	85	85
**Proposed** **Study**		Proposed DNN R3_R5_R35 Model	**95**	**93**

## Data Availability

Publicly available datasets were analyzed in this study. Ultrasound images dataset: https://www.data-in-brief.com/article/S2352-3409(19)31218-1/fulltext (accessed on 1 September 2022). Breast Histopathology Images: https://www.kaggle.com/datasets/paultimothymooney/breast-histopathology-images (accessed on 1 September 2022.).

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
