# Peer review of "A New Deep-Learning-Based Model for Breast Cancer Diagnosis from Medical Images"

_diagnostics, 2023, doi:10.3390/diagnostics13111944_

Round 1

Reviewer 1 Report

The manuscript proposes a new deep learning-based model for breast cancer diagnosis from medical images. The proposed model was compared with several deep models and existing works in the field. According to the authors' findings, it can be said that the proposed approach achieved high accuracy, precision, recall, and F1 score.

* The manuscript has typos. Please check all and fix all.

* There is a lot of research in the literature about similar topics. 

* Please emphasize the advantages and disadvantages of the proposed method in detail. 

* Please give information about the processing time for the proposed algorithm.

* Check lines 50-55, [15] is missing. 

* Check lines 61-63, [17] is missing. After line 55, the authors must change all reference numbers.

* References [19] and [20] are the same source.

* Reference [33] is not related to the topic. "...for automated breast ultrasound image classification [24-37]."

* Figures can not be seen clearly. Almost all figures have the same problem.

* Please discuss table 3 in detail.

* Authors wrote, "the Cairo University ultrasound images dataset of 210 benign and 210 malignant with the 441 associated ground truth data". Is the dataset enough for training? Is there an ethical approval document for the use of this data?

* The conclusion part should be revised.

The authors must check for typos, grammar mistakes, and parts that do not fit the journal template.

Author Response

Point 1: The manuscript has typos. Please check all and fix all.

Response 1: Thank you for your comment. We carefully reviewed the paper once again, and we also utilized Grammarly and ChatGPT software to ensure the highest quality of our writing.

Point 2: There is a lot of research in the literature about similar topics.

Response 2: We improved the related work section by adding 15 new works.

 Point 3: Please emphasize the advantages and disadvantages of the proposed method in detail.

Response 3: Corrected as instructed and the proposed method's advantages (i.e., findings) and disadvantages (i.e., limitations) have been added on page 20, lines 643 to 678, as follows:

A short review above, we can conclude the main findings of this paper as follows:

1.            Granular computing technique used in this paper by breaking down images into smaller, more granular components, can effectively extract features from images, allowing for more accurate and efficient image analysis. This leads to increased efficiency by reducing the computational complexity of image analysis tasks. Moreover, breaking down images into smaller, granular computing can improve the accuracy of image analysis tasks, leading to more reliable results.

2.            Activation functions with learnable parameters offer greater flexibility and adaptability compared to traditional activation functions with fixed parameters. This allows the network to better adapt to different types of data and tasks. These functions can also improve the flow of gradients through the network during training, making it easier to optimize the network and reduce the risk of vanishing gradients. Better regularization is another advantage of these functions. Learnable activation functions can be used as a form of regularization, helping to prevent overfitting by constraining the network's capacity and reducing the risk of memorization.

3.            In this study, a range of filters and kernels of varying sizes were employed to effectively identify patterns at multiple scales within an image. By incorporating multiple kernels within a filter, the network was able to learn diverse and robust feature representations, which helped to reduce overfitting and improve the generalization of the model. This approach enabled the model to consider a wider range of input features, leading to higher accuracy in complex tasks compared with a model that employs a single filter and kernel. The use of multiple kernels within a filter, therefore, represents an effective strategy for improving the ability of a neural network to generalize to new images, by facilitating the learning of more sophisticated features across a range of spatial scales.

4.            Utilizing a wide and depth network, shortcut connections, attention layers, auxiliary classifiers, and using a learnable activation function improve the accuracy of diagnosis and consequence decrease the load of doctors. Also, using 1x1 convolutions reduce time consumption in the model. Compared to existing breast cancer methods, the proposed model achieves the highest diagnostic accuracy.

There are two potential limitations to the presented model:

1.            The extraction of some patterns from the image may be dependent on the granularity size. In the proposed granulation, the granularity size was set to 32x32 pixels, regardless of the image size. Consequently, some patterns may not be extracted, weakening the effectiveness of granulation. However, the model's overall performance demonstrates that the proposed granulation method outperformed state-of-the-art models for the datasets under consideration.

2.            Incorporating granularity in a model requires additional time before the training process can commence. It is worth noting, however, that once these granules have been established, they can be reused multiple times.

Point 4: Please give information about the processing time for the proposed algorithm.

Response 4: The information about the processing time for the proposed algorithm has been added on page 20, lines 641- 642, as follows:

Training the model, on ultrasound and histopathology images, takes less than two hours and less than six hours, respectively, which is good compared to other DNN models.

Point 5: * Check lines 50-55, [15] is missing.

* Check lines 61-63, [17] is missing. After line 55, the authors must change all reference numbers.

* References [19] and [20] are the same source.

* Reference [33] is not related to the topic. "...for automated breast ultrasound image classification [24-37].

Response 5: Thank you so much for your attention. Corrected as instructed; we reviewed all references from scratch.

Point 6: Figures can not be seen clearly. Almost all figures have the same problem.

Response 6: We improved the resolution of all figures, however, because Figure 3 is large and needs to fit on one page, it was not possible to increase the resolution further.

Point 7: Please discuss table 3 in detail.

Response 7: Corrected as instructed and explained on page 17, lines 594 to 600, as follows:

As we used some of the features of the state-of-the-art GoogLeNet and ResNet architectures in the design of the new architecture, we compared the proposed architecture, i.e., DNN R3_R5_R35, with these architectures in Table 3. To perform this comparison, we have utilized GoogLeNet with 22 layers and ResNet with 50 layers for comparison. The results indicate that ResNet outperformed GoogLeNet in two critical evaluation metrics: Accuracy and F1 score. Inspection of the table containing the results reveals that the proposed method has surpassed both of these models and yielded a higher detection accuracy than these two state-of-the-art architectures.

Point 8: Authors wrote, "the Cairo University ultrasound images dataset of 210 benign and 210 malignant with the 441 associated ground truth data". Is the dataset enough for training? Is there an ethical approval document for the use of this data?

Response 8: Thank you for your comment.

  • We have used data augmentation techniques to increase the amount of the data. Also, we proposed a granular computing method that highlights the main features of the image before feeding it to the model. This technique makes the proposed model reach good accuracy with less data.
  • This dataset is publicly available and can be accessed (https://www.data-in-brief.com/article/S2352-3409(19)31218-1/fulltext).

Point 9: The conclusion part should be revised.

Response 9: Corrected as instructed; the conclusion section was revised and also several future works have been added.

By automating the diagnosis process, healthcare professionals can focus on providing personalized treatment plans for patients. This not only improves patient outcomes but also reduces healthcare costs by minimizing unnecessary procedures and tests. The use of machine learning and deep learning techniques in breast cancer detection has the potential to revolutionize the way breast cancer is diagnosed and managed. Incorporating these tools into healthcare systems has the potential to lower cancer-related mortality rates and enhance the overall management of breast cancer. This paper proposed a novel granular computing-based deep learning model for breast cancer detection, which is evaluated under ultrasound images and breast histopathology images datasets. The proposed model has used some effective features of GoogLeNet and Resnet architectures (such as wide and depth modules, 1x1 convolutional filters, auxiliary classifiers, and skip connection), and has added some new features such as granular computing, activation functions with learnable parameters, and attention layer to the new architecture. Granular computing can extract the important features of the image and create a new image with the important features highlighted before sending it to the training process. This feature makes the model require fewer images than in the case where granularity is not used. The proposed model achieved an accuracy improvement compared to state-of-the-art models. In particular, the deep learning model based on granular computing exhibited an accuracy of 93% and 95% on two real-world datasets ultrasound images and breast histopathology images, respectively. The model delivered promising results on the datasets. The findings may encourage radiologists and physicians to leverage the model in the early detection of breast cancer, leading to improved diagnosis accuracy, reduced time consumption, and eased workload of doctors. It has been confirmed that granular computing has a positive effect on the performance of problems where the number of available images is small, such as breast cancer.

For future work, the following items can be done: (1) The framework can be further optimized for real-time performance. Taking inspiration from the MobileNet architecture, e.g., separable con-volutions feature, the number of parameters of the proposed architecture can be reduced so that the accuracy does not decrease much. (2) Instead of granular computing, one can use fuzzy clustering. This method uses fuzzy logic to group pixels in an image into clusters based on their similarity. (3) There are usually very few medical photos. The number of these photos can be increased with techniques like Sketch2Photo [42] before starting the learning process. This will increase the accuracy of the model. (4) Over the last thirty years, hyperspectral imagery (HSI) has gained prominence for its ability to discern anomalies from natural ground objects based on their spectral characteristics. The importance of HSI has been recognized in a variety of remote sensing applications, including but not limited to object classification, hyperspectral unmixing, anomaly detection, and change detection [43]. We can use this technique to identify breast cancer. (5) The primary challenge in content-based image retrieval (CBIR) systems is the presence of a semantic gap that must be narrowed for effective retrieval. To address this issue, various techniques, such as those outlined in [44], can be employed to incorporate semantic considerations. (6) To reduce the processing time, it is suggested to use distributed and parallel similarity retrieval techniques, such as [45], on large CT image sequences. (7) The proposed framework can be extended to include other types of cancer detection, such as lung or prostate cancer. This would enable the development of a comprehensive cancer detection system that can be integrated into existing healthcare systems.

Reviewer 2 Report

Substantial amount of Refs are required throughout document

In addition to the algorithmic description, a flowchart is required

Equations require explicit numbering

More visualisation diagrams of the architectures of the designed models are required

Figure 1 appears barely legible

A distinct pathway for further work is required

acceptable

Author Response

Point 1: Substantial amount of Refs are required throughout document.

Response 1: We would like to thank you for your comment. We reviewed the references and removed some of them and added some new references. We also developed the related work section.

Point 2: In addition to the algorithmic description, a flowchart is required.

Response 2: Corrected as instructed and a flowchart of the proposed work (Fig. 1) and a new figure describing the granular computing algorithm (Fig. 2) have been added.

Point 3: Equations require explicit numbering.

Response 3: Corrected;

Point 4: More visualisation diagrams of the architectures of the designed models are required.

Response 4: Corrected as instructed and Figures 1 and 2 were added in this regard.

Point 5: Figure 1 appears barely legible.

Response 5: Thank you for your comment. Figure 1 in the previous version and Figure 3 in this version is a large figure and we have to shrink it to fit on one page. Anyway, we increased its font size to make it look a little clearer.

Point 6: A distinct pathway for further work is required

Response 6: Corrected as instructed and provided in the "Conclusion and Future work" section on page 21, lines 703 to 720, as follows:

For future work, the following items can be done: (1) The framework can be further optimized for real-time performance. Taking inspiration from the MobileNet architecture, e.g., separable convolutions feature, the number of parameters of the proposed architecture can be reduced so that the accuracy does not decrease much. (2) Instead of granular computing, one can use fuzzy clustering. This method uses fuzzy logic to group pixels in an image into clusters based on their similarity. (3) There are usually very few medical photos. The number of these photos can be increased with techniques like Sketch2Photo [42] before starting the learning process. This will increase the accuracy of the model. (4) Over the last thirty years, hyperspectral imagery (HSI) has gained prominence for its ability to discern anomalies from natural ground objects based on their spectral characteristics. The importance of HSI has been recognized in a variety of remote sensing applications, including but not limited to object classification, hyperspectral unmixing, anomaly detection, and change detection [43]. We can use this technique to identify breast cancer. (5) The primary challenge in content-based image retrieval (CBIR) systems is the presence of a semantic gap that must be narrowed for effective retrieval. To address this issue, various techniques, such as those outlined in [44], can be employed to incorporate semantic considerations. (6) To reduce the processing time, it is suggested to use distributed and parallel similarity retrieval techniques, such as [45], on large CT image sequences. (7) The proposed framework can be extended to include other types of cancer detection, such as lung or prostate cancer. This would enable the development of a comprehensive cancer detection system that can be integrated into existing healthcare systems.

Reviewer 3 Report

The following comments should be considered in order to improve the quality of the article.

1)Conclusions section should be improved.

2)The resolution of Figure 4 should be increased.

3)Information should be given about GoogLeNet and the residual block.

4)line 211: Equations should be numbered.

5)All table formats must be the same.

6)Review Work section should be improved.

The following comments should be considered in order to improve the quality of the article.

1)Conclusions section should be improved.

2)The resolution of Figure 4 should be increased.

3)Information should be given about GoogLeNet and the residual block.

4)line 211: Equations should be numbered.

5)All table formats must be the same.

6)Review Work section should be improved.

Author Response

Point 1: Conclusions section should be improved.

Response 1: Corrected as instructed; this section has been rewritten and several future works have also been added as follows:

By automating the diagnosis process, healthcare professionals can focus on providing personalized treatment plans for patients. This not only improves patient outcomes but also reduces healthcare costs by minimizing unnecessary procedures and tests. The use of machine learning and deep learning techniques in breast cancer detection has the potential to revolutionize the way breast cancer is diagnosed and managed. Incorporating these tools into healthcare systems has the potential to lower cancer-related mortality rates and enhance the overall management of breast cancer. This paper proposed a novel granular computing-based deep learning model for breast cancer detection, which is evaluated under ultrasound images and breast histopathology images datasets. The proposed model has used some effective features of GoogLeNet and Resnet architectures (such as wide and depth modules, 1x1 convolutional filters, auxiliary classifiers, and skip connection), and has added some new features such as granular computing, activation functions with learnable parameters, and attention layer to the new architecture. Granular computing can extract the important features of the image and create a new image with the important features highlighted before sending it to the training process. This feature makes the model require fewer images than in the case where granularity is not used. The proposed model achieved an accuracy improvement compared to state-of-the-art models. In particular, the deep learning model based on granular computing exhibited an accuracy of 93% and 95% on two real-world datasets ultrasound images and breast histopathology images, respectively. The model delivered promising results on the datasets. The findings may encourage radiologists and physicians to leverage the model in the early detection of breast cancer, leading to improved diagnosis accuracy, reduced time consumption, and eased workload of doctors. It has been confirmed that granular computing has a positive effect on the performance of problems where the number of available images is small, such as breast cancer.

For future work, the following items can be done: (1) The framework can be further optimized for real-time performance. Taking inspiration from the MobileNet architecture, e.g., separable con-volutions feature, the number of parameters of the proposed architecture can be reduced so that the accuracy does not decrease much. (2) Instead of granular computing, one can use fuzzy clustering. This method uses fuzzy logic to group pixels in an image into clusters based on their similarity. (3) There are usually very few medical photos. The number of these photos can be increased with techniques like Sketch2Photo [42] before starting the learning process. This will increase the accuracy of the model. (4) Over the last thirty years, hyperspectral imagery (HSI) has gained prominence for its ability to discern anomalies from natural ground objects based on their spectral characteristics. The importance of HSI has been recognized in a variety of remote sensing applications, including but not limited to object classification, hyperspectral unmixing, anomaly detection, and change detection [43]. We can use this technique to identify breast cancer. (5) The primary challenge in content-based image retrieval (CBIR) systems is the presence of a semantic gap that must be narrowed for effective retrieval. To address this issue, various techniques, such as those outlined in [44], can be employed to incorporate semantic considerations. (6) To reduce the processing time, it is suggested to use distributed and parallel similarity retrieval techniques, such as [45], on large CT image sequences. (7) The proposed framework can be extended to include other types of cancer detection, such as lung or prostate cancer. This would enable the development of a comprehensive cancer detection system that can be integrated into existing healthcare systems.

Point 2: The resolution of Figure 4 should be increased.

Response 2: This figure was created automatically by Python and therefore we could not increase its quality. Because this figure was another way of showing X-module, we deleted it.

Point 3: Information should be given about GoogLeNet and the residual block.

Response 3: We would like to thank you for your comment. Corrected as instructed and explained on page 5, section 3, lines 207 to 221, as follows:

Inspired by GoogLeNet [34] and residual block [35], and adding several other features, in this paper we developed in new deep architecture for breast cancer detection from images. GoogLeNet and residual block are based on convolutional neural network (CNN) architecture. GoogLeNet is a deep convolutional neural network architecture developed by Google's research team in 2014. It was the winner of the ImageNet Large Scale Visual Recognition Challenge (ILSVRC) in 2014 and achieved state-of-the-art performance on a variety of computer vision tasks.

The GoogLeNet architecture consists of a 22-layer deep neural network with a unique "Inception" module that enables the network to efficiently capture spatial features at different scales using parallel convolutional layers. The Inception module combines multiple convolutional filters of different sizes and concatenates their outputs, allowing the network to capture both fine-grained and high-level features. On the other hand, Residual block is a building block used in deep neural networks that helps to address the problem of vanishing gradients during training. A residual block consists of two or more stacked convolutional layers followed by a shortcut connection that bypasses these layers. The shortcut connection allows the gradient to be directly propagated to earlier layers, allowing for better optimization and deeper architectures.

Point 4: line 211: Equations should be numbered.

Response 4: Corrected;

Point 5: All table formats must be the same.

Response 5: Thank you. Corrected; in the current version of the article, all the tables related to the results have the same format.

Point 6: Review Work section should be improved.

Response 6: Thank you for your comment. Corrected as instructed; we have added several new works (15 new works) to the related work section (page 2, section 2).

Reviewer 4 Report

·        The authors should rewrite the abstract to follow this structure: background, objective, materials and methods, results, conclusion, and recommendations.

·        Check lines 46 and 47; let them align with the other sentences. The way they are now shows they are starting a paragraph wrong, so please correct it.

·        Reference 17 was omitted and not cited in the body of the paper. Check and cite where appropriate.

·        Check line 109; citing more than five papers in a sentence should be avoided because personally engaging in that reduces the clarity of the sentence, relevance, and impact of the citation, so please avoid it.

·        Line 147, you said three key ways…. But you outlined four, so correct the statement.

·        Part of Figure 1 has been cut off; this is not professional, so correct this mistake.

·        The authors should discuss the dataset used for the implementation. What are the attributes of the dataset, how was the dataset split, into what ratio, etc?

·        The authors should state how the study performance was accessed or evaluated.

·        How did the authors tune the optimal hyperparameter of all models? It should be described clearly.

·        How did you solve the problem of overfitting and small dataset

·        Overall, the English language and presentation style should be improved significantly. There were a lot of grammatical errors and typos. I suggest you have a colleague proficient in English and familiar with the subject matter review your manuscript or contact a professional editing service.

·        The limitation of the study should be stated, and they should present future research work.

·        The study should be compared with existing systems (state-of-the-art).

·        Source codes should be provided for replicating the study.

·        We are in May 2023, and I could see only 1 2022 citation and 4 2021 citations. Citing recent literature have several advantages for both authors and journal. It can assist authors in establishing their credibility, demonstrating the relevance of their research and help to avoid plagiarism. In the same way assist journals in increasing their visibility, improving their reputation, increase their citation rates, and meeting reader expectations. Hence, for this reason I have suggested some recent literature that you are to cite and reference in your article.

a.      Xiong, S., Li, B., & Zhu, S. (2022). DCGNN is a single-stage 3D object detection network based on density clustering and graph neural networks. Complex & Intelligent Systems. doi: 10.1007/s40747-022-00926-z

b.      Liu, H., Xu, Y., & Chen, F. (2023). Sketch2Photo: Synthesizing photo-realistic images from sketches via global contexts. Engineering Applications of Artificial Intelligence, 117, 105608. doi: https://doi.org/10.1016/j.engappai.2022.105608

c.      Feng, H., Yang, B., Wang, J., Liu, M., Yin, L., Zheng, W.,... Liu, C. (2023). Identifying Malignant Breast Ultrasound Images Using ViT-Patch. Applied Sciences, 13(6), 3489. doi: 10.3390/app13063489

d.      Li, Z.B. et al., Integrating of lipophilic platinum(IV) prodrug into liposomes for cancer therapy on patient-derived xenograft model. CHINESE CHEMICAL LETTERS, 2022. 33(4): p. 1875-1879.

e.      Song, X.Y., et al., Construction of a biotin-targeting drug delivery system and its near-infrared theranostic fluorescent probe for real-time image-guided therapy of lung cancer. CHINESE CHEMICAL LETTERS, 2022. 33(3): p. 1567-1571.

f.       Lu, L., Dong, J., Liu, Y., Qian, Y., Zhang, G., Zhou, W., Zhao, A., Ji, G., & Xu, H. (2022). New insights into natural products that target the gut microbiota: Effects on the prevention and treatment of colorectal cancer. Frontiers in pharmacology, 13, 964793. https://doi.org/10.3389/fphar.2022.964793

g.      Li, B., Wang, W., Zhao, L., Yan, D., Li, X., Gao, Q.,... Liao, Y. (2023). Multifunctional AIE Nanosphere-Based “Nanobomb” for Trimodal Imaging-Guided Photothermal/Photodynamic/Pharmacological Therapy of Drug-Resistant Bacterial Infections. ACS Nano, 17(5), 4601-4618. doi: 10.1021/acsnano.2c10694

h.      "Zhang, P.; Sun, B., A comparative study on the curative effect of Da Vinci robotic surgery system and conventional surgery in the treatment of gynecological tumors, EUROPEAN JOURNAL OF GYNAECOLOGICAL ONCOLOGY 2022, 43(5), 114."

i.       Wang, S., Hu, X., Sun, J., & Liu, J. (2023). Hyperspectral anomaly detection using ensemble and robust collaborative representation. Information Sciences, 624, 748-760. doi: https://doi.org/10.1016/j.ins.2022.12.096

j.       Zhuang, Y., Chen, S., Jiang, N., & Hu, H. (2022). An Effective WSSENet-Based Similarity Retrieval Method of Large Lung CT Image Databases. KSII Transactions on Internet & Information Systems, 16(7). doi: 10.3837/tiis.2022.07.013

k.      Zhuang, Y., Jiang, N., Xu, Y., Xiangjie, K., & Kong, X. (2022). Progressive Distributed and Parallel Similarity Retrieval of Large CT Image Sequences in Mobile Telemedicine Networks. Wireless communications and mobile computing, 2022. doi: 10.1155/2022/6458350

·        Overall, the English language and presentation style should be improved significantly. There were a lot of grammatical errors and typos. I suggest you have a colleague proficient in English and familiar with the subject matter review your manuscript or contact a professional editing service.

Author Response

Point 1: The authors should rewrite the abstract to follow this structure: background, objective, materials and methods, results, conclusion, and recommendations.

Response 1: Corrected as instructed and the abstract has been rewritten.

 Point 2: Check lines 46 and 47; let them align with the other sentences. The way they are now shows they are starting a paragraph wrong, so please correct it.

Response 2: Thank you for your comment; corrected it as instructed.

Point 3: Reference 17 was omitted and not cited in the body of the paper. Check and cite where appropriate.

Check line 109; citing more than five papers in a sentence should be avoided because personally engaging in that reduces the clarity of the sentence, relevance, and impact of the citation, so please avoid it.

Response 3: We would like to thank you for your comment. Corrected as instructed; we reviewed all references from scratch.

Point 4: Line 147, you said three key ways…. But you outlined four, so correct the statement.

Response 4: Thank you. Corrected; thank you for your attention.

Point 5: Part of Figure 1 has been cut off; this is not professional, so correct this mistake.

Response 5: We did not understand exactly what you meant. Anyway, we made changes to the figure and also tried to improve its quality to some extent.

Point 6: The authors should discuss the dataset used for the implementation. What are the attributes of the dataset, how was the dataset split, into what ratio, etc?

Response 6: Regarding the Cairo University ultrasound images dataset, we added the following explanation on page 15, lines 548 to 561.

The Cairo University ultrasound images dataset was collected in 2018 that consists of 780 images with an average image size of 500 × 500 pixels [36]. The images are categorized into three classes, which are normal (133 images), benign (487 images), and malignant (210 images). The data collected at baseline include breast ultrasound images among women in ages between 25 and 75 years old. The number of patients is 600 female patients. Because the number of images in different classes is unbalanced, this may cause the model to learn some classes better than others, and this can cause the model to perform inappropriately during use or testing. To prevent this from happening, we randomly selected an equal number of images from each class. Before starting the training process of the model, due to the lack of data, we start the data augmentation process. We resize the dataset using cubic interpolation to fit the input requirements of the model. For augmentation, we applied width and height shifts of 0.1 and a horizontal flip, which triple the size of the dataset. With this technique, we tripled the number of data for each class. After this process, we split the dataset into training and test sets, allocating approximately 80% for training and 20% for testing. Of course, we have not used these new images for testing. We maintain the sequence of each image so that every image appears only once in each of the aforementioned sets.

Regarding the breast histopathology images dataset, we added the following explanation on page 19, lines 623 to 629, as follows:

We also applied the proposed model to the breast histopathology images dataset to further evaluate it. The original dataset consisted of 162 whole-mount slide images of Breast Cancer specimens scanned at 40x. From that, 277,524 patches of size 50x50 were extracted (198,738 IDC negative and 78,786 IDC positive). Invasive Ductal Carcinoma (IDC) is the most common subtype of all breast cancers. We choose 40,000 images in total from the dataset: 20,000 random images from both classes. We split the dataset into training and test sets, allocating 80% for training and 20% for testing.

Point 7: The authors should state how the study performance was accessed or evaluated.

Response 7: Thank you for your comment. We have added the evaluation metrics on page 13, lines 514 to 530.

Point 8: How did the authors tune the optimal hyperparameter of all models? It should be described clearly.

Response 8: We have added the following explanation in this regard on page 13, section 4 as follows:

To train the model, the following settings were employed: Adam optimizer with a learning rate set to 0.0001, 100 epochs, a batch size of 32, and a dropout rate of 50%. The number of epochs in a CNN is one of the hyperparameters that can be tuned to improve the performance of the model. The number of epochs refers to the number of times the entire training dataset is passed through the CNN during the training phase. To tune the number of epochs, we used early stopping technique. Early stopping is a method used to prevent overfitting by monitoring the validation loss during training and stopping the training process once the validation loss starts to increase. After applying this technique we set the number of epochs to 100.

We have utilized various filters and kernels (kernel size). By specifying multiple values for the kernel parameter within a filter, our model can effectively identify patterns that occur at different scales within an image. The incorporation of multiple kernels also assists in reducing overfitting and improving the generalization of the model. This is because including filters with varying kernel sizes compels the network to learn more diverse and robust feature representations, leading to an improved ability to generalize the model to new images.

Point 9: How did you solve the problem of overfitting and small dataset.

Response 9: Thank you for your comment. We used several techniques in the paper for avoiding overfitting, for example, early stopping, learnable activation functions, and short connections.

The second dataset used in this paper is not small and has a large number of images. For the first dataset, we also used data augmentation techniques to increase the amount of the data.

For both datasets, we used a granular computing method that highlights the main features of the image before feeding it to the model. This technique makes the proposed model reach good accuracy with less data.

Point 10: Overall, the English language and presentation style should be improved significantly. There were a lot of grammatical errors and typos. I suggest you have a colleague proficient in English and familiar with the subject matter review your manuscript or contact a professional editing service.

Response 10: Thank you for your comment. We carefully reviewed the article once again, and we also utilized Grammarly and ChatGPT software to ensure the highest quality of our writing.

Point 11: The limitation of the study should be stated, and they should present future research work.

Response 11: Corrected as instructed and explained on page 21, lines 670 to 678, as follows:

There are two potential limitations to the presented model:

1.            The extraction of some patterns from the image may be dependent on the granularity size. In the proposed granulation, the granularity size was set to 32x32 pixels, regardless of the image size. Consequently, some patterns may not be extracted, weakening the effectiveness of granulation. However, the model's overall performance demonstrates that the proposed granulation method outperformed state-of-the-art models for the datasets under consideration.

2.            Incorporating granularity in a model requires additional time before the training process can commence. It is worth noting, however, that once these granules have been established, they can be reused multiple times.

Point 12: The study should be compared with existing systems (state-of-the-art).

Response 12: Corrected as instructed; Table 5 has been updated in this regard. Two recent publications from 2023 have been incorporated into Table 5. The updated table now shows a comprehensive comparison of the proposed model against 15 existing models. Furthermore, we also compared the proposed model against nine state-of-the-art models, as depicted in Table 4.

Point 13: Source codes should be provided for replicating the study.

Response 13: To address this comment, we added Data Availability Statement and Code Availability Statement to the end of the paper before the references.

Point 14: We are in May 2023, and I could see only 1 2022 citation and 4 2021 citations. Citing recent literature have several advantages for both authors and journal. It can assist authors in establishing their credibility, demonstrating the relevance of their research and help to avoid plagiarism. In the same way assist journals in increasing their visibility, improving their reputation, increase their citation rates, and meeting reader expectations. Hence, for this reason I have suggested some recent literature that you are to cite and reference in your article.

  1. Xiong, S., Li, B., & Zhu, S. (2022). DCGNN is a single-stage 3D object detection network based on density clustering and graph neural networks. Complex & Intelligent Systems. doi: 10.1007/s40747-022-00926-z

  1. Liu, H., Xu, Y., & Chen, F. (2023). Sketch2Photo: Synthesizing photo-realistic images from sketches via global contexts. Engineering Applications of Artificial Intelligence, 117, 105608. doi: https://doi.org/10.1016/j.engappai.2022.105608

  1. Feng, H., Yang, B., Wang, J., Liu, M., Yin, L., Zheng, W.,... Liu, C. (2023). Identifying Malignant Breast Ultrasound Images Using ViT-Patch. Applied Sciences, 13(6), 3489. doi: 10.3390/app13063489

  1. Li, Z.B. et al., Integrating of lipophilic platinum(IV) prodrug into liposomes for cancer therapy on patient-derived xenograft model. CHINESE CHEMICAL LETTERS, 2022. 33(4): p. 1875-1879.

  1. Song, X.Y., et al., Construction of a biotin-targeting drug delivery system and its near-infrared theranostic fluorescent probe for real-time image-guided therapy of lung cancer. CHINESE CHEMICAL LETTERS, 2022. 33(3): p. 1567-1571.

  1. Lu, L., Dong, J., Liu, Y., Qian, Y., Zhang, G., Zhou, W., Zhao, A., Ji, G., & Xu, H. (2022). New insights into natural products that target the gut microbiota: Effects on the prevention and treatment of colorectal cancer. Frontiers in pharmacology, 13, 964793. https://doi.org/10.3389/fphar.2022.964793

  1. Li, B., Wang, W., Zhao, L., Yan, D., Li, X., Gao, Q.,... Liao, Y. (2023). Multifunctional AIE Nanosphere-Based “Nanobomb” for Trimodal Imaging-Guided Photothermal/Photodynamic/Pharmacological Therapy of Drug-Resistant Bacterial Infections. ACS Nano, 17(5), 4601-4618. doi: 10.1021/acsnano.2c10694

  1. "Zhang, P.; Sun, B., A comparative study on the curative effect of Da Vinci robotic surgery system and conventional surgery in the treatment of gynecological tumors, EUROPEAN JOURNAL OF GYNAECOLOGICAL ONCOLOGY 2022, 43(5), 114."

  1. Wang, S., Hu, X., Sun, J., & Liu, J. (2023). Hyperspectral anomaly detection using ensemble and robust collaborative representation. Information Sciences, 624, 748-760. doi: https://doi.org/10.1016/j.ins.2022.12.096

  1. Zhuang, Y., Chen, S., Jiang, N., & Hu, H. (2022). An Effective WSSENet-Based Similarity Retrieval Method of Large Lung CT Image Databases. KSII Transactions on Internet & Information Systems, 16(7). doi: 10.3837/tiis.2022.07.013

  1. Zhuang, Y., Jiang, N., Xu, Y., Xiangjie, K., & Kong, X. (2022). Progressive Distributed and Parallel Similarity Retrieval of Large CT Image Sequences in Mobile Telemedicine Networks. Wireless communications and mobile computing, 2022. doi: 10.1155/2022/6458350

Response 14: Thank you so much for your constructive comment. Regarding your comment, we reviewed the references suggested by you and referred five of those (b, c, i, j, and k) that were most related to our work. In addition to them, we also added some new references.

Round 2

Reviewer 1 Report

I checked previous and revised versions of the submitted manuscript. I still have the same decision. There is a lot of similar research in the literature. The manuscript doesn't have significant outcomes. It is no different from other studies in the literature and is a repetition of similar things.

I mentioned the lack of the dataset in my previous review (Point 8). The authors added the following part to the manuscript "In the study presented in [17], various deep-learning models were employed to classify breast cancer ultrasound images based on their benign, malignant, or normal status. A dataset comprising a total of 780 images was utilized, and data augmentation and preprocessing techniques were applied. Three 117 models were evaluated for classification. Specifically, ResNet50 achieved an accuracy of 85.4%, ResNeXt50 achieved 85.83%, and VGG16 achieved 81.11%. "

There are still unclear points in the manuscript. What kind of data augmentation techniques were used? Why weren't given the information about it? 

It would be better if the authors could submit the manuscript to one of the machine-learning conferences. I recommend that the authors revise the study according to the feedback from the conference, update the dataset, and send it to the journal, again.

It must be improved.

Reviewer 2 Report

Thanks 

Reviewer 4 Report

All of my comments have been attended to as requested.

The English quality is very minor.